# COVID-19 Patients’ Medication Management during Transition of Care from Hospital to Virtual Care: A Cross-Sectional Survey and Audit

**DOI:** 10.3390/pharmacy11050157

**Published:** 2023-09-28

**Authors:** H. Laetitia Hattingh, Catherine Edmunds, Saberina Buksh, Sean Cronin, Brigid M. Gillespie

**Affiliations:** 1Allied Health Research, Gold Coast Health, Gold Coast, QLD 4215, Australia; 2School of Pharmacy and Medical Sciences, Griffith University, Gold Coast, QLD 4222, Australia; 3School of Pharmacy, the University of Queensland, Brisbane, QLD 4102, Australia; 4Medical Services, Clinical Governance and Research, Gold Coast Health, Gold Coast, QLD 4215, Australia; catherine.edmunds@health.qld.gov.au; 5Pharmacy Department, Gold Coast Health, Gold Coast, QLD 4215, Australia; saberina.buksh@health.qld.gov.au (S.B.); sean.cronin@health.qld.gov.au (S.C.); 6NHMRC Wiser Wounds CRE, MHIQ, Griffith University, Gold Coast, QLD 4222, Australia; b.gillespie@griffith.edu.au; 7Nursing and Midwifery Education and Research Unit, Gold Coast Health, Gold Coast, QLD 4215, Australia

**Keywords:** virtual care, medication management, pharmacist review

## Abstract

Background: Virtual models of care were implemented to ease hospital bed pressure during COVID-19. We evaluated the medication management of COVID-19 patients transferred to virtual models of care. Method: A retrospective audit of COVID-19 patients transferred from inpatient units to virtual care during January 2022 and surveys from patients transferred during December 2021–February 2022 was carried out. Results: One hundred patients were randomly selected: mean age 59 years (SD: 19.8), mean number of medicines at admission 4.3 (SD: 4.03), mean length of virtual ward stay 4.4 days (SD: 2.1). Pharmacists reviewed 43% (43/100) of patients’ medications during their hospital stay and provided 29% (29/100) with discharge medicine lists at transfer. Ninety-two (92%) patients were prescribed at least one new high-risk medicine whilst in hospital, but this was not a factor considered to receive a pharmacist medication review. Forty patients (40%) were discharged on newly commenced high-risk medicines, and this was also not a factor in receiving a pharmacist discharge medication list. In total, 25% of patient surveys (96/378) were returned: 70% (66/96) reported adequate medicine information at transfer and 55% (52/96) during the virtual model period. Conclusion: Patient survey data show overall positive experiences of medication management and support. Audit data highlight gaps in medication management during the transfer to a virtual model, highlighting the need for patient prioritisation.

## 1. Introduction

The number of patients infected with COVID-19 resulted in an influx to hospitals, with health systems facing unprecedented strain [1]. Hospitals worldwide had to rapidly reorientate health services and implement new models of care to cope with increased service demands [2,3]. These models include emergency department avoidance for high-risk patients, e.g., cancer patients and patients with cardiovascular disease [4,5,6,7,8]. Other measures implemented to build the capacity of health systems included opening respiratory clinics, expanding emergency departments and increasing the availability of general hospital beds in public and private hospitals [9]. Health services also implemented collaborative patient self-management and remote patient monitoring programs through the implementation of digital health solutions such as video consultations, telemedicine [10,11,12,13,14,15] and Hospital-at-Home models [16].

Australia followed a quarantine and isolation model to avoid the spread of the virus [9], and interstate borders were restricted until most of the population was double vaccinated [17]. However, the opening of the Queensland borders in December 2021 and the rise in the COVID-19 Omicron variant caused a surge of the virus in South-East Queensland. It was expected to reach a peak toward the end of January 2022, and in preparation for this surge, the Gold Coast Hospital and Health Service implemented several new and innovative virtual models of care. These included a virtual ward and a Hospital-In-The-Hotel service. These innovative virtual models were implemented in a staged approach throughout December 2021 and mid-February 2022 as hospital demands changed.

The isolation requirements for COVID-19 and the nature of these virtual models of care posed unique medication management and safety challenges. These included the inability of hospital pharmacists to provide face-to-face medicine reconciliation, review services when patients were admitted, or medication counselling services at the point of transfer, as pharmacists were not allowed to be physically present at inpatient units. Research indicates the value of inpatient unit pharmacists having access to medication histories during hospital admission [18,19], conducting clinical reviews [20] and counselling patients during discharge from the hospital [21,22,23]. Systematic reviews highlight the number of discrepancies in medical errors at the point of discharge and the need for medication reconciliation by pharmacists, which is an important strategy to avoid discrepancies [24,25]. Therefore, the medication management and safety impact of pharmacists not being present in inpatient units poses challenges. Pharmacists had to rely on surrogate markers to identify patients for medication review, such as patients on polypharmacy (≥5 medicines). Pharmacists could only communicate with patients via the telephone to explore their health literacy and compliance with taking medicines. Also, patients in virtual models of care had to self-administer their medicines, similar to other Hospital-In-The-Home programs; however, the identification of medication-related problems relied on telephone interactions with patients. This is different from other Hospital-In-The-Home programs that incorporate regular face-to-face interactions, which facilitate observation of the patient’s environment and visual compliance (e.g., pill counts) [26,27].

Studies have evaluated virtual medication management services, such as telehealth consultations [28,29,30] and community pharmacies supporting hospitals for patients’ medication management [31,32]. However, there is a gap in the literature on the medication management and safety aspects that are unique to hospital virtual models of care.

The aim of this study was to evaluate the medication management and risk factors of COVID-19 patients who were transferred from the hospital to virtual models of care. Specific objectives were to:Evaluate patients’ potential medication-related risk factors;Explore patients’ medication management during transfer whilst in a virtual model of care;Obtain patients’ perspectives on the administration and management of their medicines whilst in a virtual model of care.

For this study, medication management services incorporated pharmacist medication reviews during an inpatient unit stay and the provision of discharge medication lists at transfer to virtual care. Medication safety aspects focused on patients’ risk factors for potential medication-related harm (MRH) and being prescribed a new high-risk medicine.

## 2. Methods

This was an observational study incorporating a cross-sectional survey and retrospective audit to evaluate the effect of an intervention, namely the transfer of patients from an inpatient unit to virtual models of care. The Strengthening the Reporting of Observational Studies in Epidemiology (STROBE) guidelines [33] were followed throughout this review. This study was conducted in accordance with the provisions of the Declaration of Helsinki and the Australian National Health and Medical Research Council *National Statement on Ethical Conduct in Human Research (2023)*. Ethics approval was received by the Gold Coast Hospital and Health Service (GCHHS) Human Research Ethics Committee (Ref No. LNR2020QGC83951).

### 2.1. Study Sites/Settings

This study was conducted at GCHHS and included COVID-19 patients transferred from inpatient units at Gold Coast University and Robina Hospitals to virtual models of care (virtual ward or Hospital-In-The-Hotel) from December 2021 until mid-February 2022. At the time of the pandemic surge, both hospitals experienced staff shortages due to COVID-19.

Virtual models involved the transfer of COVID-19 patients from inpatient units to the community with hospital support with patients classified as virtual inpatients. Patients qualified for this service if they required ongoing low-level care or minimal follow-up during the completion of their virtual isolation period. To limit the spread of COVID-19 infection, pharmacists were not present at COVID-19 inpatient units and had to provide medication history and review services remotely over the phone for admitted inpatients whilst these patients were physically at the hospital. Pharmacists provided medication counselling over the phone when patients were transferred to a virtual model.

### 2.2. Data Sources

The evaluation involved (1) a retrospective audit of patients’ medical records and (2) a patient survey conducted via a telephone or online (Figure 1):

1.
*Patients’ medical records*


A list of all patients who were transferred from an inpatient unit to a virtual model of care throughout January 2022 was extracted from the local electronic medical record system to conduct a retrospective audit during March 2022. Paediatric and maternity patients and patients admitted as inpatients for less than 24 h prior to their transfer to a virtual model were excluded.

Data included details of the inpatient unit patients were transferred from, the date of transfer, length of stay (LoS) in the inpatient unit and length of duration in a virtual model of care. Further data collected from patients’ medical records included:Patients’ demographic details (age, gender), including criteria recorded for their referral to virtual care;Risk factors associated with MRH: vision impairment, mobility/dexterity impairment/frail/pressure injury risk, fall risk, communication difficulty, cognitive impairment/dementia/mental health issues, and stroke [34,35,36];Medication details on admission to the hospital and transfer to a virtual model;If patients were prescribed a new high-risk medicine (i.e., was not prescribed the medicine prior to admission) during hospital admission or a transfer to a virtual model according to the APINCH classification system: anti-infectives, injections of potassium and other electrolytes, insulin, narcotics and other sedatives, chemotherapeutic agents, heparin and anticoagulants [37];Medication management services provided by pharmacists:◦Pharmacist review during hospital admission;◦Discharge medication record when transferred to a virtual model;Readmission within 30 days.

2.
*Patient surveys*


COVID-19 adult patients transferred from any inpatient unit to a virtual model of care between 9 December 2021 and 10 February 2022 were invited to participate in a survey to obtain their perspectives on the management and self-administration of their medicines, including their confidence to administer their own medicines and self-reported adherence.

Survey questions were developed specifically for this study. The literature on patient interviews/surveys were considered [38,39], the Queensland Health Patient Reported Experience Survey–Care for COVID-19 [40] and a validated tool to explore patients’ self-reported adherence and confidence in the use of their medicines (MARS) [41] were used to support survey development. The 19-question survey included five-point Likert-scale questions (e.g., not at all confident–extremely confident, not at all concerned–extremely concerned), ‘yes’, ‘no, and ‘unsure’ questions, and open-ended questions to comment on the service (Appendix A). The face validity of the draft survey was assessed by a researcher, two pharmacists and two consumers and feedback was incorporated. The completion of the survey was estimated to take approximately five minutes.

### 2.3. Recruitment

Patients transferred from an inpatient unit to a virtual model between 1 and 10 February 2022 were telephoned toward the end of February 2022. Information about the study was provided over the phone; there was an opportunity for questions and patients to provide verbal consent before responding to survey questions. As these patients were phoned, verbal consent was considered the most practical form of consent, with the date and time of consent recorded in an Excel spreadsheet. This was approved by the GCHHS Human Research Ethics Committee (Ref No. LNR2020QGC83951). Patients who were transferred from an inpatient unit to a virtual model between mid-December 2021 and the end of January 2022 were invited to complete an online Microsoft Forms survey via a letter mailed to them and a mobile phone text message. They were requested to complete the survey once only. The letter included a QR code with a link to the online MS Forms survey. The text message similarly had a link to the survey. The mailout occurred during April 2022, and the text messages were sent out on 21 April and was repeated on 6 June 2022.

### 2.4. Data Analysis

Audit and survey data were entered into Microsoft Excel© spreadsheets and survey data into Stata 17 (Stat Corp., College Station, Tx, USA) for analysis. Descriptive statistics summarised relative and absolute frequencies. Fisher’s exact test was used to test associations between patients’ MRH risk factors and those prescribed a new high-risk medicine with the provision of pharmacist medication management services (pharmacist review and discharge medication records). A *p*-value of <0.05 indicated statistical significance.

## 3. Results

A local GCHHS COVID-19 dashboard with data extracted from the electronic medical record system showed that 378 patients were transferred from inpatient units to a virtual model (virtual ward or Hospital-In-The-Hotel) between 9 December 2021 and 10 February 2022.

### 3.1. Audit of Medical Records

After exclusions were applied, 100 patients were randomly selected from a list of 187 patients (53.5%) who were transferred from inpatient units to a virtual model between 1 January and 31 January 2022 (Table 1). Patients’ mean age was 58.8 years (SD: 19.8), the mean LoS in hospital was 3.0 days (SD: 1.7) and mean LoS in the virtual model was 4.4 days (SD: 2.1). Patients were on a mean of 4.3 medicines at admission to hospital (SD: 4.0) and pharmacists reviewed 43% (43/100) of patients’ medicines during their hospital stay and provided discharge medicine lists for 29% (29/100) of patients during their transfer to a virtual model. Forty-two patients (42%) had at least one risk factor for medication-related harm. Ninety-two patients (92%) were prescribed and administered at least one new high-risk medicine whilst in hospital, with 25% (25/100) of patients prescribed two high-risk medicines. Most of the patients (65%; 65/100) had a carer whilst in virtual care; however, 12% (12/100) lived alone, and 8% (8/100) had to care for someone else. Most patients completed virtual care at home and 14% (14/100) at a hotel (Hospital-In-The-Hotel). Twenty-six patients were readmitted to the hospital within four weeks, and one patient died following discharge. None of the primary reasons for readmission were recorded as medicine-related.

Regarding the risk factors for potential MRH, 32% (32/100) of patients had either mobility issues, dexterity impairment were frail, or had a pressure injury risk, and 15% (15/100) of patients had a risk of falling. Fisher’s exact test showed no association between patients’ risk factors and receiving an inpatient unit pharmacist review or a discharge medication list at transfer (Table 2).

Table 3 summarises patients who were prescribed a new high-risk medicine during hospital admission or at transfer to a virtual model. In terms of the types of high-risk medicines prescribed, 80% (80/100) received anticoagulants, 47% (47/100) received anti-infectives, and 36% (36/100) received narcotics/sedatives whilst in hospital. Of the patients on anti-infectives, 20% (20/100) received one and 12% and 11% (12/100 and 11/100) were on two or three anti-infectives, respectively. Fisher’s exact test showed no association between patients prescribed a new high-risk medicine during hospital admission and receiving a pharmacist medication review. Forty patients (40%) received a prescription for a new high-risk medicine at the point of transfer, of which 27% (27/100) were prescribed a new anti-infective and 11% (11/100) narcotics/sedatives. There was no association between being prescribed a new high-risk medicine at transfer and receiving a discharge medication list prepared by a pharmacist.

### 3.2. Patient Surveys

A total of 96/378 surveys (25.4%) were received, comprising 67.9% (19/28) telephone and 22.0% (77/350) online surveys (Table 4). Overall, 35.1% (33/94) of patients responded that they were not contacted by hospital staff about their medicines whilst being in a virtual model of care, but responses showed that 70.2% (66/94) patients reported receiving adequate medicine information at transfer, and 55.3% (52/94) during the virtual model period. Likert-scale responses showed that patients were moderately confident (28.6%; 26/91) or extremely confident (56.0%; 51/91) in how to use their medicines whilst in the virtual model of care.

Fifty-three participants provided open-ended responses to questions on how their medicines were managed whilst being an inpatient and under virtual care. The majority of patients provided positive comments, and those that raised concerns mostly related to uncertainty on how to manage their medicines once in virtual care:

“Medicine was left in a bag outside room door when left hospital and only instructions were on containers” P4

“I was concerned about getting Clexane^®^ [subcutaneous anticoagulant].”P13

“While in the hospital I felt very supported and my medication was discussed with me by the Dr via phone on day one. He also supplied my daily medication to ensure I had my own supply for when I left quarantine. When transferred there was a lot of confusion and there were days I wasn’t contacted at all and days I was contacted by both the physical ward and virtual ward. Medications were not mentioned again.” P37

## 4. Discussion

Virtual models of care are innovative approaches introduced to ease hospital bed pressure and facilitate the appropriate supported care of large numbers of patients requiring low-level care during a COVID-19 surge. There were 378 patients transferred from an inpatient unit to a virtual model of care between 9 December 2021 and 10 February 2022 at GCHHS to release beds to patients who required a high level of care provided by hospital facilities. Patients in virtual models of care were still classified as hospital inpatients but were managed remotely by hospital staff. Of specific interest is the mismatch between the audit data and patients’ survey data: the audit data highlighted a lack of prioritisation of patients at risk of potential MRH, whereas the survey data showed that patients felt they received sufficient medication management support.

The surveyed patients reported a positive experience with the transition of care from the hospital and the virtual care they received. A Sydney, New South Wales study of 265 patients in home isolation and hotel quarantine similarly found that patients responded well to virtual care in a pandemic context, independent of whether they were at home or at a hotel [42]. However, the Sydney study incorporated video consultations, and one of the patients commented: “The video conference allows a face-to-face interaction, so much better than a mere phone call. Words combined with facial expressions are so much better” [42]. A study conducted in Scotland compared follow-up video consultations with face-to-face and telephone consultations. The outcomes showed that patients liked video consultations, although technical problems were commonly reported by clinicians and patients, and the authors recommended addressing these before promoting the uptake of video consultations [43]. There is a need to compare telephone versus video consultations for patients in virtual models of care in terms of both patient and clinician preferences and medication management safety and outcomes. The COVID-19 pandemic is referred to as health care’s digital revolution [44], as health services internationally had to adapt to the increased use of technology to manage large numbers of patients [45].

A study conducted in Spain evaluated the outcomes of 63 COVID-19 patients who were transferred to a hospital through a home model and found that the model was safe and an efficacious alternative to hospitalisation [46]. However, their model included daily medical and nurse visits as well as tests conducted at home. The models of care evaluated throughout our study did not incorporate face-to-face home visits but rather phone calls and, in that sense, were unique. Open-ended responses in the survey showed that some patients were confused about their medicine administration. For example, patients who had to continue using subcutaneous anticoagulants were concerned about the administration of their medicine.

Virtual models of care not only release bed pressure but also facilitate physical distancing between patients and healthcare providers [47,48]. However, careful planning is needed to consider patients’ medication management and safety risks during their transfer to a virtual model of care, as transitions of care are high-risk periods for medication-related harm (MRH) [49,50,51,52,53]. A systematic review of the international literature showed that 17–51% of older people experience MRH within 30 days of hospital discharge [53], which is estimated to be 37% at 60 days post-discharge [54]. From an Australian perspective, approximately 250,000 (19%) Australian hospital admissions occur annually due to a medication-related event [55]. Transfer to a virtual model of care qualifies as a transition of care and requires quality handover and documentation for primary care healthcare providers to undertake ongoing care of patients. The audit of medical records showed no associations between those who were on a newly prescribed high-risk medicine at transfer and those who received a pharmacist discharge medication list. This poses a medication safety risk. Indeed, various studies have shown the value of pharmacists’ involvement in discharge medicine reconciliation when reducing medication errors [21,56,57].

The absence of a pharmacist discharge medicine list could have affected the quality of communication with general practitioners, as discharge medicine lists are incorporated into discharge summaries. The lack of pharmacist involvement at the point of transfer could have also impacted on patients’ compliance with newly prescribed medicines as research shows that patients often overestimate the likelihood of adverse effects [58].

Due to isolation requirements during the initial surge of the COVID-19 pandemic in Queensland, inpatient pharmacists were not able to provide face-to-face care and had to provide medication reviews and counselling services remotely via telephone. Research has shown that face-to-face medicine counselling results in improved medication adherence and persistency compared to the dispensing of medicines only [59]. A study conducted in the Netherlands showed that both patients and healthcare professionals preferred face-to-face consultations over telemedicine for new patients, e.g., hospital discharge patients [60]. Future transfer to virtual care models should explore various telehealth models, e.g., video counselling, which could incorporate the use of tools such as pictograms and smartphone programs, as these have been proven effective in supporting patients’ understanding and subsequent medication adherence [61]. The lack of pharmacist face-to-face contact with patients may also have negatively impacted on the prioritisation of patients for medication management services.

The effectiveness of pharmacy services may have been impacted by organisational challenges such as low staffing levels and poor communication between ward and pharmacy staff. A survey of Australian hospital pharmacists found that tele-pharmacy services introduced as a result of the COVID-19 pandemic disrupted pharmacists’ workflow and increased their workload [62]. The results of this study identified a need to develop future virtual models of care that incorporate the prioritisation of patients for pharmacist medication services.

A strength of this study was the use of two data sources to obtain insights from different angles: patients’ medical records and a patient survey. Audit data comprised more than 50% of patients who complied with inclusion criteria. However, the survey response was low, which limited the generalisability of the data. Survey links were sent out via text message as well as mailouts. While patients were asked to complete it once only, some patients could have completed it multiple times. Although data were collected on patients who were readmitted to the hospital following discharge, the data could not be evaluated in terms of potential medication-related causes.

## 5. Conclusions

The surge of the COVID-19 pandemic in Queensland at the end of 2021/beginning of 2022 placed unprecedented pressure on hospitals. Virtual models of care introduced innovative approaches to transfer low-risk patients from the hospital to be managed remotely at home or in a hotel to increase inpatient bed capacity. An evaluation of virtual models of care suggests the need to prioritise patients at potential risk of medication-related harm for pharmacist medication management services. Insights from this study could be used to guide the implementation of similar models of care for future COVID-19 waves or pandemics.

## Figures and Tables

**Figure 1 pharmacy-11-00157-f001:**
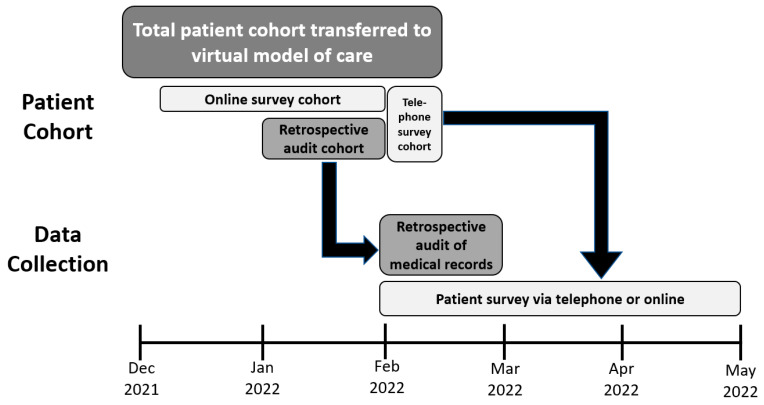
Overview of patient cohorts and data collection.

**Table 1 pharmacy-11-00157-t001:** Summary of patient details (*n* = 100).

Variables	Mean	SD
**Age** (years)	58.8	19.8
**Length of stay**	**Mean**	**SD**
In hospital (days)	3.0	1.7
In virtual ward (days)	4.4	2.1
**Virtual model of care**		** *n* ** **/%**
Virtual ward		86
Hospital-In-The-Hotel		14
**Number of patients with risk factors for medication-related harm (some had multiple)**		** *n* ** **/%**
Vision impairment		3
Mobility/Dexterity impairment/Frail/Pressure injury risk		32
Fall risk		15
Communication difficulty		6
Cognitive impairment/Dementia/Mental health issues		5
**Number of patients with risk factors for medication-related harm**		** *n* ** **/%**
No risk factors		58
One risk factor		25
Two risk factors		15
Three risk factors		2
**Medication use**	**Mean**	**SD**
Number of medicines at admission per patient	4.3	4.0
Number of new high-risk medicine in hospital per patient	2.5	1.6
Number of new high-risk medicine in virtual model of care per patient	0.5	0.7
**Patients on new high-risk medicines whilst in hospital**		** *n* ** **/%**
No high-risk medicines		8
One high-risk medicine		22
Two high-risk medicines		25
Three high-risk medicines		19
Four high-risk medicines		12
Five high-risk medicines		7
Six high-risk medicines		7
**Type of high-risk medicine (some on multiple)**		
Anti-infectives		47
Injections of potassium and other electrolytes		13
Insulin		13
Narcotics/sedatives		36
Chemotherapeutic agents		0
Heparin and other anticoagulants		80
**Patients on new high-risk medicines whilst in virtual model**		** *n* ** **/%**
No high-risk medicines		60
One high-risk medicine		31
Two high-risk medicines		7
Three high-risk medicines		2
**Type of high-risk medicine (some on multiple**)		
Anti-infectives		27
Injections of potassium and other electrolytes		0
Insulin		4
Narcotics/sedatives		11
Chemotherapeutic agents		0
Heparin and other anticoagulants		4
**Medication management services**		
Received pharmacist review		43
Received discharge medicine list		29
Received discharge summary		48

**Table 2 pharmacy-11-00157-t002:** Patients’ potential risk factors associated with medication-related harm and those who received a pharmacist review or discharge medication list (*n* = 100).

Risk Factor for Medication-Related Harm	No. with Risk Factor	Received Pharmacist Review (%)			Received Pharmacist Discharge Medication List (%)		
		*n* (%)	95% Confidence Interval	* *p*-Value	*n* (%)	95% Confidence Interval	* *p*-Value
Vision impairment	3	0	-	0.257	0	-	0.554
Mobility/Dexterity impairment/Frail/Pressure injury risk	32	14 (43.8%)	0.273; 0.617	1.000	9 (28.1%)	0.149; 0.466	1.000
Fall risk	15	6 (40.0%)	0.177; 0.674	1.000	3 (20.0%)	0.059; 0.500	0.543
Communication difficulty	6	2 (33.3%)	0.051; 0.822	0.697	2 (33.3%)	0.051; 0.822	1.000
Cognitive impairment/Dementia/Mental health issues	5	2 (40.0%)	0.050; 0.894	1.000	2 (40.0%)	0.050; 0.894	0.626
Stroke	3	2 (66.7%)	0.010; 0.997	0.576	1 (33.3%)	0.003; 0.990	1.000
Any risk factor	42	17 (40.5%	0.265; 0.562	0.688	12 (28.6%)	0.167; 0.444	1.000

* Fisher’s exact test.

**Table 3 pharmacy-11-00157-t003:** Patients prescribed new high-risk medicine(s) during inpatient stay and during transfer to virtual model while receiving pharmacist review or discharge medication list (*n* = 100).

New High-Risk Medicine (HRM)	Hospital Admission/Inpatient Unit Stay				Transfer to Virtual Model			
	No. on New HRM	No. Received Pharmacist Medication Review (%)	95% Confidence Interval	* *p*-Value	No. on New HRM	No. Received Pharmacist Discharge Medication List (%)	95% Confidence Interval	* *p*-Value
Anti-infectives	47	23 (48.9%)	0.348; 0.633	0.313	27	11 (40.7%)	0.235; 0.606	0.139
Injections of potassium and other electrolytes	13	8 (61.5%)	0.316; 0.847	0.229	0	0	-	-
Insulin	13	7 (53.9%)	0.258; 0.797	0.550	4	0	-	0.320
Narcotics/sedatives	36	15 (41.7%)	0.264; 0.587	1.000	11	3 (27.3%)	0.766; 0.629	1.000
Chemotherapeutic agents	0	0	-	-	0	0	-	-
Heparin and other anticoagulants	80	34 (42.5%)	0.320; 0.537	1.000	4	2 (50%)	0.040; 0.060	0.578
Any high-risk medicine	92	41 (44.6%)	0.346; 0.549	0.460	40	14 (35.0%)	0.216; 0.513	0.369

* Fisher’s exact test.

**Table 4 pharmacy-11-00157-t004:** Summary of patients’ survey responses.

Question	Yes		No		Unsure	
	*n*	%	*n*	%	*n*	%
Did one of the staff members talk to you about how to use your medicines when you left the hospital ward? (*n* = 95)	51	53.7	39	41.1	5	5.3
Did you receive a written list of your medicines with instructions on how to use them when you left the hospital ward? (*n* = 95)	37	39.0	49	51.6	9	9.5
Did your medicines change whilst you were in hospital? (*n* = 95)	35	36.8	49	51.6	11	11.6
Do you believe that the information you received from the hospital about your medicines when you were transferred was adequate for you to know how to take your medicine? (*n* = 94)	66	70.2	19	20.2	9	9.6
Did you have enough supply of your medicines whilst at home or hotel? (*n* = 94)	67	71.3	18	19.2	9	9.6
Did you receive advice about how to take your medicines while at home or in the hotel? (*n* = 94)	52	55.3	33	35.1	9	9.6
Did you have the opportunity to ask questions about your medicines whilst at home or in the hotel? (*n* = 92)	47	51.1	34	37.0	11	12.0
Do you have a special way to help you remember to take your medicines? (*n* = 92)	29	31.5	54	58.7	9	9.8
Did you miss any medicine doses whilst at home or in the hotel? (*n* = 92)	8	8.7	77	83.7	7	7.6
Did you know how to contact a hospital pharmacist for assistance while you were at home or in the hotel if you needed support? (*n* = 92)	49	53.3	36	39.1	7	7.6
Was there a time you wanted to speak to a hospital pharmacist about your medicines while you were at home or in the hotel, but failed to make contact? (*n* = 92)	6	6.5	76	82.6	10	10.9

## Data Availability

The data presented in this study are available on request from the corresponding author. The data are not publicly available due to the possibility of patients being reidentified.

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
