# Peer review of "COVID-19 Patients’ Medication Management during Transition of Care from Hospital to Virtual Care: A Cross-Sectional Survey and Audit"

_pharmacy, 2023, doi:10.3390/pharmacy11050157_

Round 1
Reviewer 1 Report
Dear authors,
I would like to express my appreciation for reviewing this article. The structure of the paper is excellent, and the chosen topic is both interesting and relevant. Below, you will find my suggestions and questions for improving the text:
METHODS
You followed the STROBE guidelines meticulously during the writing process, as it has been done very rigorously. However, it is challenging to adhere to the methodology, and it occupies almost the entire paper. For instance, Figure 1 is unclear in terms of the lower part of the image - I suggest visually connecting the upper and lower parts of the image with arrows. What is crucial to clarify is whether the patients whose medical records were analyzed are the same as those for whom the study of experience and satisfaction with virtual care was conducted. If they are not, does it make sense to correlate conclusions about patient satisfaction with poorer audit data? This needs to be discussed in the paper.
Furthermore, it is crucial to describe the criteria that were applied by the pharmacists during the hospital stay and the transition to virtual care when choosing a patient needing medication review or discharge list.
Patient survey
The methods section described how a specific questionnaire for the research was created, which also included the MARS. Why haven't you reported on adherence in the results and discussed it?
RESULTS
At the outset, I would like to have a table of general patient information included in the analysis of medical records. It is essential to provide a more detailed description of these patients, including gender, age, therapy during their stay, the number of medications, and other clinical data.
Tables 1 and 2 are very confusing in the light of lack of a general table of patient data.
The table lists risk factors in a sample of 100 patients, and the text states that the numbers correspond to the frequency observed, for example, 15 patients, or 15%, had a risk of falling. However, in data presented in this manner, it is not clear how many patients in total had risk factors and which had one versus multiple risk factors.
From the data presented in this manner, it is impossible to conclude the accuracy of the statistical test mentioned in Table 1 because it is impossible to verify or construct a contingency table from the provided data. Please clarify this.
The same issue appears in Table 2. Once again, it's unclear how many patients have multiple HRMs and how many have only one. How was the contingency table created? I suggest that in Tables 1 and 2, the risk factors and HRMs are described in detail. In my opinion, it would make more sense to then have a separate table that clearly and unambiguously presents the Fisher's test for the group that received medication review or discharge list as a form of care about the presence of any HRM or risk factor, rather than for each HRM or risk factor individually.
DISCUSSION:
It would be valuable to discuss the results after the changes.
Best regards
Author Response
We have addressed all of the issues, please refer to the attached table

Reviewer 2 Report
Thank you very much for giving an opportunity to review the present manuscript.
The authors evaluated the medication management of COVID-19 patients transferred to virtual models of care. This is a problem that requires solutions around the world, and the implementation of new models of care were described in detail in this paper. Although the topic of this research is timely and important, there are some concerns that should be made before publication.
[Materials and Methods]
The survey questions were designed for this study with reference to literature previously described. Do you evaluate the results of the questionnaire survey using scoring? As shown in Table 3, the response rate for each item is important, but I think scoring will provide a more scientific opinion.
[Results]
Why did you randomly select 100 patients?
[Table 1 and 2]
For each item “Received pharmacist review” and “Received pharmacist discharge medication list”, it would be better to list the value, percentage, and 95% confidence interval instead of just listing the p-value.
Author Response
We have addressed all of the issues, please refer to the table atatched

Round 2
Reviewer 1 Report
Thank you for implementing the sugestions.
Reviewer 2 Report
The paper has been appropriately revised according to the reviewers' comments and deserves acceptance.